# Transcriptional Responses of Stress-Related Genes in Pale Chub (*Zacco platypus*) Inhabiting Different Aquatic Environments: Application for Biomonitoring Aquatic Ecosystems

**DOI:** 10.3390/ijerph191811471

**Published:** 2022-09-12

**Authors:** Won-Seok Kim, Kiyun Park, Jae-Won Park, Sun-Ho Lee, Ji-Hoon Kim, Yong-Jun Kim, Gun-Hee Oh, Bong-Soon Ko, Ji-Won Park, Cheol Hong, Tae-Sik Yu, Ihn-Sil Kwak

**Affiliations:** 1Department of Ocean Integrated Science, Chonnam National University, Yeosu 59626, Korea; 2Fisheries Science Institute, Chonnam National University, Yeosu 59626, Korea

**Keywords:** *Zacco platypus*, stress-related genes, transcriptional expression, integrated biomarker response, lake environment

## Abstract

Pale chub (*Zacco platypus*) is a dominant species in urban rivers and reservoirs, and it is used as an indicator to monitor the effects of environmental contaminants. Gene responses at the molecular level can reflect the health of fish challenged with environmental stressors. The objective of this study was to identify correlations between water quality factors and the expression of stress-related genes in *Z. platypus* from different lake environments (Singal and Juam Lakes). To do so, transcriptional responses of genes involving cellular homeostasis (heat-shock protein 70, *HSP70*; heat-shock protein 90, *HSP90*), metal detoxification (metallothionein, *MT*), and antioxidation (superoxide dismutase, *SOD*; catalase, *CAT*) were analyzed in the gill and liver tissues of *Z. platypus*. *HSP70*, *HSP90*, and *MT* genes were overall upregulated in *Z. platypus* from Singal Lake, which suffered from poorer water quality than Juam Lake. In addition, gene responses were significantly higher in Singal Lake outflow. Upregulation of *HSP70*, *HSP90*, and *MT* was significantly higher in *Z. platypus* gills than in the liver tissue. In addition, integrated biomarker response and heatmap analysis determined correlations between expression of biomarker genes or water quality factors and sampling sites of both lakes. These results suggest that stress-related genes used as multiple biomarkers may reflect spatial characteristics and water quality of different lake environments, and they can be used for biomonitoring and ecological risk assessment.

## 1. Introduction

Aquatic environments are the ultimate sink for a wide variety of pollutants whose range and intensity are increasing in the organisms inhabiting these environments [1]. Scientific technological development and industrial advancement have resulted in a massive upsurge in the amount of industrial chemicals produced in recent decades. As the release of wastewater chemicals and manmade industrial products increases, emerging and persistent pollutants are frequently detected in aquatic ecosystems. These processes also lead to the enrichment of toxic and bioaccumulative compounds in aquatic organisms, such as fish [2]. Fish can be a useful species for biomonitoring aquatic environments because of their ubiquity and key ecological position at a high tropic level in aquatic ecosystems [3,4,5]. To understand how fish respond to the stressors of an aquatic environment, the use of complementary biomarkers is recommended for risk monitoring in aquatic ecosystems [6,7,8].

From the organism to molecular level, biomarker responses have been broadly used to assess the potential effects of environmental stressors, including toxic chemicals, wastewater mixtures, and hazardous substances, in aquatic environments [5,7,9,10,11]. Biomarkers as “early warning” signs have been used to identify the comprehensive effects of toxicants on the health of the individual before these effects are observed at the population level in an ecosystem [12,13]. In addition, biomonitoring through a comprehensive understanding of biomarkers can represent measures of health in the freshwater environments. In particular, molecular biomarkers can reflect potential toxicities in the form of altered gene expression in cellular defense systems in fish [5,7,14,15,16]. The pale chub (*Zacco platypus*) is a dominant species in freshwater environments that is distributed in a wide range of eastern Asian regions such as Korea, Japan, and China [17,18]. It is omnivorous and consumes aquatic insects and algae as main food sources [5]. At this time, the molecular responses in fish have mostly been studied with respect to the effects of exposures to chemical compounds such as benzo(a)pyrene in *Oryzias latipes* or *Oreochromis niloticus* fish [4,16]. The previous reported studies on molecular biomarkers were mostly limited to toxicity tests in the laboratory, using experimental rearing species that are relatively easy to handle. In addition, although antioxidant and physiological responses to toxicants or wastewater are reported in *Z. platypus*, there is not enough information regarding the molecular responses to environmental stressors in wild fish such as *Z. platypus*.

Ecosystems of standing (lentic) reservoirs easily accumulate environmental pollutants from natural and anthropogenic sources [19,20]. In this study, we analyzed stress-related genes in *Z. platypus* inhabiting two different lentic ecosystems (Singal and Juam Lakes) in order to evaluate gene expression responses for biomonitoring applications in aquatic environments. Singal Lake is situated in an urban environment, which exposes it to anthropogenic inputs, whereas Juam Lake is surrounded by forest. We observed the transcription of genes involving cellular homeostasis (heat-shock protein 70, *HSP70*; heat-shock protein 90, *HSP90*), metal detoxification (metallothionein, *MT*), and antioxidation (superoxide dismutase, *SOD*; catalase, *CAT*) in the gills and livers of *Z. platypus* from each lake. In addition, the integration of biomarker responses (IBR) and heatmap analysis were used to compare gene expression, water quality factors, and sampling sites (inflow or outflow) of each lake.

## 2. Materials and Methods

### 2.1. Sampling Preparation

Fish were collected at 2–3 sites within the two major lakes (Singal and Juam Lakes) of South Korea (Figure 1) using a kick net (4 mm × 4 mm, 30 min), cast net (6 mm × 6 mm, 10 times per site), gill net (100 m total length, 1.5 m height, 45 mm and 12 mm stretched mesh sizes), and fyke net (three pockets, 3 mm mesh size, 20 × 2.4 m lead height). Fish were sampled in September 2020, and tissue samples were immediately extracted from fish (*n* = 3 at each site) in the field and stored in RNAlater (Invitrogen, Waltham, MA, USA). The fish had an average weight of 11.25 ± 4.41 g, total length of 10.59 ± 1.13 cm, and body length of 8.58 ± 0.84 cm. The samples were transported to the laboratory and stored at 4 °C for 12 h, before being stored at −80 °C.

### 2.2. Total RNA Extraction and cDNA Synthesis

Total RNA was extracted using RNA isoplus (Takara, Shiga, Japan) according to the manufacturer’s protocol. Recombinant DNase I (Takara, Shiga, Japan) treatment was used to remove genomic DNA contamination in the extracted RNA. RNA concentration and quality (260:280 ratios >1.8) were verified using a microplate reader with a nanodrop plate (Thermo Scientific, Waltham, MA, USA). RNA integrity was confirmed by 1.2% agarose gel electrophoresis. Then, total RNA (2 µg) was used as a template to synthesize cDNA using a PrimerScriptTM First-Strand cDNA Synthesis Kit (Takara, Shiga, Japan) according to the manufacturer’s protocol. cDNA was stored at −20 °C, in 15-fold diluted conditions.

### 2.3. Gene Expression Analysis Using Quantitative Real-Time PCR (RT-qPCR)

To amplify the expression of stress-related genes in fish tissue, RT-qPCR was performed on a CFX Connect Real-Time PCR system (Bio-Rad, Hercules, CA, USA). Before the experiment, gene-specific primers were designed on the basis of the full or partial coding sequences of candidate genes using Primer3 software (Version 0.4.0). An efficiency test was conducted to validate the designed primers (E: 88–96%). Table 1 contains the information for all primers used in this study. Each reaction was conducted in a final volume of 20 µL containing 10 µL of Accuprep 2× Greenstar qPCR Master Mix (Bioneer, Daejeon, Korea), 6 µL of DEPC-treated water, 0.5 µL of each forward and reverse primer (10 pM), and 3 µL of 15-fold-dilution cDNA as a template. RT-qPCR was carried out for 40 cycles at 95 °C for 15 s, a primer-specific temperature (Table 1) for 30 s, and 60 °C for 45 s. Melting curves were determined by increasing the temperature from 65 °C to 95 °C. All samples were amplified in triplicate to ensure reproducibility. The relative expression level of each gene was determined using glyceraldehyde-3-phosphate dehydrogenase (GAPDH) as an internal reference gene and calculated using the 2^−ΔΔCt^ method [21].

### 2.4. Integration of Biomarker Responses (IBR)

The IBR index was calculated according to Beliaeff and Burgeot [22]. Briefly, biomarkers (*HSP70*, *HSP90*, *SOD*, *CAT*, and *MT* expression) were standardized to compare the relative values at each site as follows:$$Y=\frac{\left(X-m\right)}{s},$$
where *Y* is the standardized biomarker response, *X* is the general mean value of each biomarker, and *s* is the standard deviation of *X*. The score (*S*) was computed as *S* = *Y* + |min|, where *S* ≥ 0, and |min| is the absolute value of each biomarker value. The score was averaged at different levels of biomarker responses to create a star plot by connecting endpoints of each vector using R (version 4.0.5). The IBR value can be calculated follows:$$A_{i}=\frac{S_{i}}{2}\beta \left(S_{i}\cos\beta +S_{i+1}\sin\beta \right),$$
where
$$\beta =\mathrm{Arctan}\left(\frac{S_{i+1}\sin\alpha }{S_{i}-S_{i+1}\cos\alpha }\right),$$
where *α* = 2π/*n* radians, *Sn* + 1 = *S*1, and *n* = 3.

### 2.5. Statistical Analysis

R statistical software (version 4.0.5) was used for statistical analyses in this study. Data are presented as the mean ± standard deviation. A one-way analysis of variance was conducted to test for significant differences in biomarker responses among different survey areas. In addition, significant differences were determined using a Tukey test at the *p* < 0.05 (*) and *p* < 0.01 (**) significance levels.

## 3. Results

### 3.1. Water Quality and Hydrological Environments of Lakes

Hydrological and geographical features differ between Singal and Juam Lakes of South Korea. Singal Lake is located in an urban area, close to a factory, whereas Juam Lake is situated in a forest (Figure 1). Juam Lake is much larger and deeper than Singal Lake (Table 2). The physicochemical properties of lake water were analyzed at inflow, middle, and outflow sites of Singal Lake, and at middle and outflow sites of Juam Lake. Water temperatures were higher in Juam Lake than Singal Lake. Dissolved oxygen (DO) and pH were lower in Juam Lake than in Singal Lake, possibly due to differences in water temperature and depth. Sampling sites within Singal Lake had electric conductivity (EC) values ranging from 273 to 723 µS/cm (Table 2), while Juam Lake had an average EC of 74.4 µS/cm. Moreover, compared to Juam Lake, Singal Lake had higher values of most water quality factors, including levels of organic contaminants (chemical oxygen demand, COD), nutrient levels (total organic carbon, TOC and total nitrogen, TN), nitrogen levels (NO_3_-N, NH_3_-N), an indicator of phytoplankton biomass (chlorophyll *a*, Chl-*a*), and dissolved TOC (DTOC).

### 3.2. Transcriptional Responses of Stress-Related Genes in Z. platypus

#### 3.2.1. Cellular Homeostasis in *Zacco platypus* from Different Lakes

The gene expression of heat-shock proteins (*HSP70* and *HSP90*) in gills (Figure 2) and livers (Figure 3) of *Z. platypus* was analyzed to evaluate the cellular homeostasis responses in fish collected from Singal and Juam Lakes. In *Z. platypus* gills, *HSP70* expression was only significantly higher in the outflow of Singal Lake (71.6-fold) and in the outflow of Juam Lake (14.2-fold) (Figure 2A). *HSP90* expression tended to increase from the inflow (16.0-fold) to outflow (26.0-fold) site in Singal Lake, but the opposite pattern was observed in Juam Lake (Figure 2B). In livers, *HSP70* mRNA expression was highest in *Z. platypus* collected from the middle of Juam Lake and did not reach significance at other survey points (Figure 3A). In addition, *HSP90* expression was similar to that of *HSP70*, although *HSP90* levels were lower than *HSP70* levels in liver tissue. There were no significant levels of *HSP90* expression in Singal and Juam Lakes (Figure 3B). The expression of HSPs was generally higher in the gill tissue than liver tissue of *Z. platypus* from both lakes (Figure 2 and Figure 3).

#### 3.2.2. Antioxidant Defense in *Zacco platypus* from Different Lakes

The transcription of antioxidant enzymes (*SOD* and *CAT*) in gills (Figure 2) and livers (Figure 3) of *Z. platypus* was analyzed to identify how responses to oxidative stress differed among fish collected from Singal and Juam Lakes. In the gills of *Z. platypus* collected from Singal Lake outflow, *SOD* expression increased significantly, mirroring *HSP90* expression patterns (Figure 2C). In addition, the level of *CAT* mRNA increased in the outflow (16.2-fold) of Singal Lake (Figure 2D). However, the highest level (32.0-fold) of *CAT* expression was observed in the middle of Juam Lake. In liver tissue, *SOD* was upregulated in the middle of Singal Lake (1.9-fold), but downregulated in Juam Lake compared to Singal Lake (Figure 3C). Furthermore, the lowest level of *CAT* transcripts was found in *Z. platypus* from the middle of Singal Lake and Juam Lake (Figure 3D). However, the expression levels of *SOD* or *CAT* were not significant in the liver of *Z. platypus* from each lake. The mRNA levels of antioxidant genes were generally higher in gill tissue than in the liver tissue of *Z. platypus* from both lakes (Figure 2 and Figure 3).

#### 3.2.3. Metal Detoxification in *Zacco platypus* from Different Lakes

*MT* expression was analyzed in the gills (Figure 2) and livers (Figure 3) of *Z. platypus* collected from Singal and Juam Lakes to investigate changes in metal detoxification in these environments. In the gills of *Z. platypus*, *MT* expression resembled that of *SOD* in Singal Lake (Figure 2E), with the expression of *MT* being significantly highest in Singal Lake outflow. In general, low *MT* expression was observed in Juam Lake (middle: 13.8-fold; outflow: 28.5-fold), with *MT* expression being non-significantly elevated in *Z. platypus* collected from the middle of Juam Lake (4.2-fold) (Figure 3E). The transcriptional levels of the metal detoxification gene considered here were generally higher in gill tissue than in the liver tissue of *Z. platypus* from both lakes (Figure 2 and Figure 3).

### 3.3. Integration of Biomarker Responses (IBR) and Heatmap Analysis

The IBR index values in each tissue were calculated to evaluate the relative environmental health of Singal and Juam Lakes using sampling sites from both lakes and biomarker gene levels from *Z. platypus* (Figure 4). A heatmap analysis was used to determine correlations between sampling sites and physicochemical factors in order to assess water quality (Figure 5). In *Z. platypus* gills, the IBR values of *HSP70*, *HSP90*, *MT*, and *SOD* were significantly highest in Singal Lake outflow, although a high *CAT* IBR was observed in Juam Lake inflow (Figure 4A). However, in *Z. platypus* livers, the IBR index values of *HSP70*, *HSP90*, and *MT* were high in the middle of Juam Lake (Figure 4B). The IBR index of antioxidant genes such as *SOD* or *CAT* was higher in liver tissue than in gills of *Z. platypus* from Singal Lake. The biological response values for HSPs and *MT* were high in *Z. platypus* gill tissue from Singal Lake outflow (IBR value = 12.62). In addition, the positive correlation between Singal Lake sampling sites and physicochemical properties indicating impaired water quality in the heatmap suggests that water quality was poorer in Singal Lake (Figure 5).

## 4. Discussion

*Zacco platypus* has been used as a model for evaluating the biological and molecular consequences of the surrounding water environment, as well as exposure to heavy metals and toxicants [5,7,8,9]. In addition, genes have been used as biomarkers that reflect aquatic environmental conditions shaped by wastewater effluents or environmental pollutants [5,7]. In addition, the molecular biomarker approach using *Z. platypus*, which is distributed over a wide range of rivers, is a useful tool that can represent the health of the fish habitat environment using native fish. This study is the first to analyze multilevel genetic biomarkers in *Z. platypus* gill and liver tissues from standing (lentic) water of different lake environments.

Fish gills are in direct contact with the surrounding environment and have several vital functions involving ionic and osmotic regulation, aquatic gas exchange, excretion of nitrogenous wastes, and acid–base regulation [23,24]. In addition, gills play pivotal roles in stress responses to environmental hypoxia [25]. In the present study, basal expression levels of biomarker genes were higher in *Z. platypus* gills than in livers of fish collected from Singal and Juam Lakes. These results indicate that fish gills are an optimal organ for the assessment of aquatic environments such as lakes, as well as rivers and streams. In *Z. platypus* gills, *HSP70* and *HSP90* were significantly upregulated in Singal Lake sampling sites, especially the outflow. HSPs play an important role in biological metabolism and cell activity, and many studies have utilized these genes as molecular markers for external stress [26]. In addition, through activities such as chaperone folding and translocation, HSPs are reported to play a role in maintaining protein homeostasis in the face of oxidative stress caused by heavy metals [27]. In this study, molecular responses against potential environmental stressors appeared higher in the gills of *Z. platypus* inhabiting Singal Lake than in those inhabiting Juam Lake. A manufacturing plant and food plant sit close to the outflow of Singal Lake (Figure 1). Moreover, the responses of HSPs were not dependent on water temperature (Juam Lake > Singal Lake) in both lakes, although HSPs act as molecular chaperones to prevent protein denaturation in response to heat-shock stress [15].

*MT* was also significantly upregulated in the gills of *Z. platypus* from Singal Lake in this study. *MT* primarily contributes to cytoprotection, which prevents oxidation stress caused by the toxicity of heavy metals, such as cadmium, zinc, and copper [28]. In addition, *MT* plays critical roles in the regulation of metal homeostasis in the intracellular detoxification of heavy metals [7]. In past studies, *MT* mRNA was significantly increased in *Z. platypus* located downstream of wastewater treatment plant effluents [7,29]. However, antioxidant enzyme responses to the production of oxidative stress differed by *Z. platypus* tissue type in this study. Upregulated expression of *SOD* was observed in the gills and liver of *Z. platypus* from Singal Lake, whereas upregulation of *CAT* differed in Singal and Juam Lakes depending on whether the gill or liver tissue was examined. When an intracellular oxidation reaction occurs in response to an externally derived xenobiotic, antioxidant genes create balance within the oxidation mechanism. Superoxide radicals (O_2_^−^) produced due to external stress are transformed by *SOD* and converted into hydrogen peroxide (H_2_O_2_) and molecular oxygen (O_2_) [30]. H_2_O_2_ can accumulate in biological cells and tissues and cause cell death. To mitigate the toxic effects of H_2_O_2_ and protect the organism from oxidative damage, *CAT* decomposes H_2_O_2_ into H_2_O and O_2_. *SOD* and *CAT* are considered to be the first line of defense against ROS damage [31]. Differing expression patterns of *SOD* and *CAT* were also reported in *Z. platypus* exposed to wastewater effluents [7]. Thus, antioxidant biomarker genes can reflect the oxidative stress conditions experienced by *Z. platypus* from Singal Lake, which has poor water quality.

The IBR index was first proposed for the purpose of star plot visualizations as an indicator of environmental stress [22]. The IBR index has been used to assess the stress of organisms in aquatic environments [5,7]. In this study, IBR index values were analyzed on the basis of the star plot areas to identify the adverse effects of different lake environments. In the gills of *Z. platypus*, the total IBR value was the highest in Singal Lake outflow (12.62), clearly suggesting that *Z. platypus* gills can reflect stressful environments. The poor water quality properties observed at this site may have been caused by an adjacent factory. IBR values also indicated stressful water conditions downstream, potentially affected by wastewater effluents [7]. The high IBR index values obtained in this study indicated that impaired water quality or exposure stressors (toxicants or high temperature) were correlated with an overall higher degree of stress [32,33,34].

## 5. Conclusions

In this study, we identified correlations between the water conditions of two lake ecosystems (Singal and Juam) and the expression of stress-related genes as biomarkers in the gills and liver of *Z. platypus*. Cellular homeostasis (*HSP70* and *HSP90*) and metal detoxification (*MT*) genes were significantly upregulated in the gills of *Z. platypus* collected from the outflow area of Singal Lake, which possessed poorer water quality conditions than Juam Lake. In *Z. platypus*, the gills proved to be a more useful organ to assess stressful water environments of lentic ecosystems than the liver. However, the expression patterns of antioxidant genes (*SOD* and *CAT*) were more sensitive in liver tissues than in the gills of *Z. platypus* from Singal Lake, although the relative levels of *SOD* and *CAT* were lower in the liver than in the gills of *Z. platypus*. The IBR value also indicated a high degree of stress, presented as a high IBR index, in *Z. platypus* gills at the Singal Lake outflow. These results indicate that transcriptional responses of biomarker genes in lake-inhabiting fish can indicate stressful water environments in ecosystems.

## Figures and Tables

**Figure 1 ijerph-19-11471-f001:**
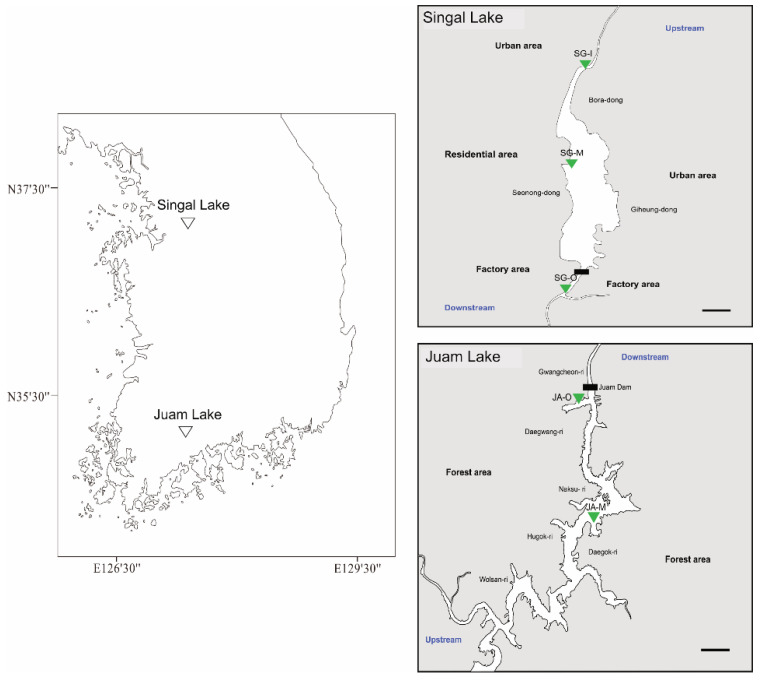
Location of the study area and sampling sites (Black line: 1 km).

**Figure 2 ijerph-19-11471-f002:**
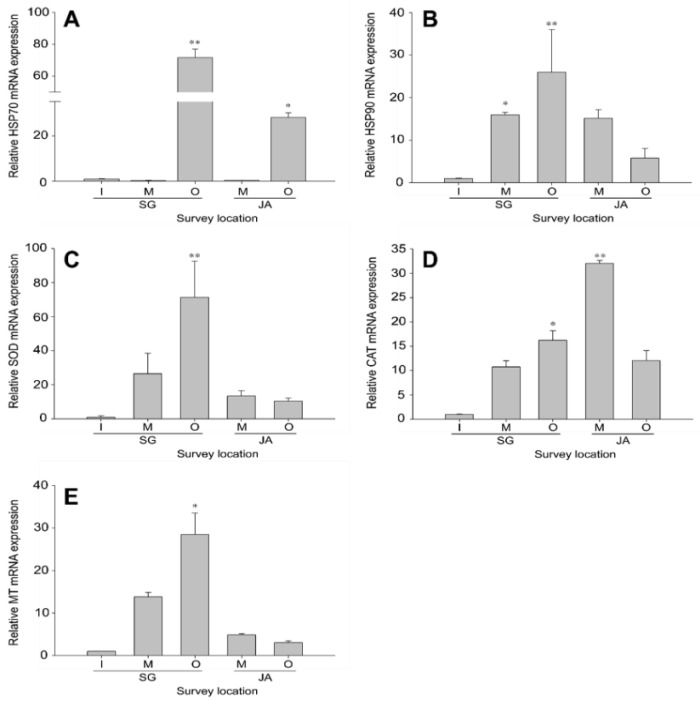
Gene expression analysis of *HSP70* (**A**), *HSP90* (**B**), *SOD* (**C**), *CAT* (**D**), and *MT* (**E**) in *Zacco platypus* gills. The data are presented as the mean ± SD. The expression levels of each gene were compared with the expression level in Singal Lake inflow (expression level = 1). Significant differences are indicated with asterisks: * *p* < 0.05 and ** *p* < 0.01.

**Figure 3 ijerph-19-11471-f003:**
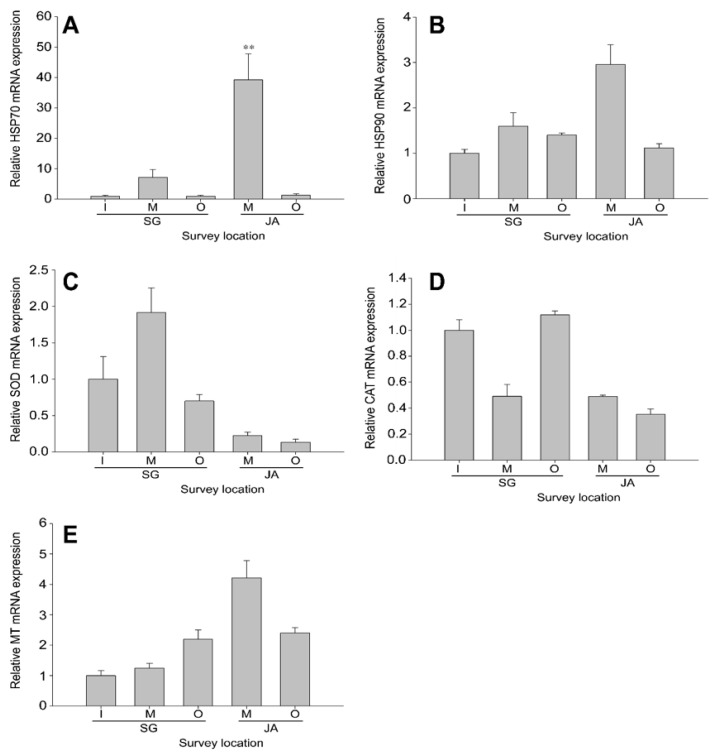
Gene expression analysis of *HSP70* (**A**), *HSP90* (**B**), *SOD* (**C**), *CAT* (**D**), and *MT* (**E**) in *Zacco platypus* liver. The data are presented as the mean ± SD. The expression levels of each gene were compared with the expression level in Singal Lake inflow (expression level = 1). Significant differences are indicated with asterisks: ** *p* < 0.01.

**Figure 4 ijerph-19-11471-f004:**
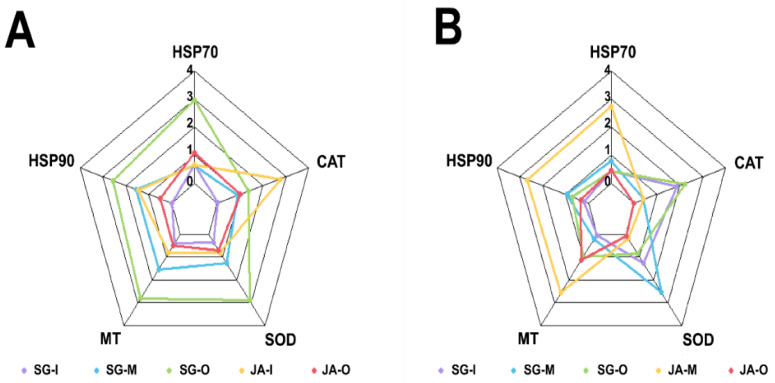
Star plots of IBR values for evaluating multiple gene responses in the gills (**A**) and liver (**B**) of *Zacco platypus* from each sampling site of Singal and Juam lakes.

**Figure 5 ijerph-19-11471-f005:**
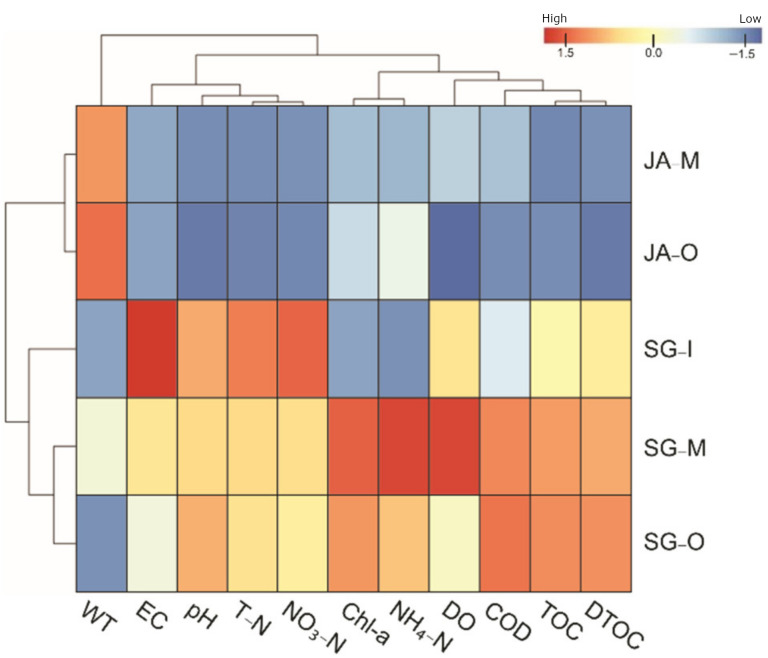
Heatmap of water quality factors at each sampling site in Singal and Juam Lakes. Cell color represents high or low levels of water quality factors. Each group underwent Z-score normalization.

**Table 1 ijerph-19-11471-t001:** List of specific primers used for quantitative real-time PCR.

Gene	Primer	Tm (°C)	Efficiency (%)	Reference
*GAPDH*	F: ACGCGGAAGGCCATACC	53	90	Kim and Jung, 2016 [7]
	R: GGCTGTGGGCAAAGTCATTC			
*HSP70*	F: CCTCAATGGTCCTGGTGAAG	61	96	KY926431
	R: TCACCTTCTGCCCCAGATAA			
*HSP90*	F: TGGTGTGGGCTTCTACTCTG	60	88	KM201321
	R: CCCTCTTCTCCTCGACGTAC			
*SOD*	F: GAAGGAGGATGACTTGGGTAAGG	62	90	KF515699
	R: CCGGCGTTGCCAGTTTTA			
*CAT*	F: AAATCCGCAGACTCACCTAAA	53	89	KF515698
	R: GGACGCAAACCCCAGAAA			
*M* *T*	F: GATTGCGCCAAGACTGGAA	60	91	KC952875
	R: CTGGCAGTTAGTGCACTTGCA			

**Table 2 ijerph-19-11471-t002:** Hydrological and physicochemical factors in each lake.

Lake	Hydrological Factors			Physicochemical Factors												
	Basin Area (km^2^)	Reservoir Area (km^2^)	Maximum Water Level (EL·m)	Place	Depth (m)	WT (°C)	Do (mg·L^−1^)	pH	EC (µmhos/cm)	COD (mg·L^−1^)	TOC (mg·L^−1^)	TN (mg·L^−1^)	Chl-a (mg·m^−3^)	NO_3_-N (mg·L^−1^)	NH_3_-N (mg·L^−1^)	DTOC (mg·L^−1^)
Singal (SG)	53	2.3	46	Inflow (I)	0.4	17.4	12.9	8.3	723.0	4.3	3.0	4.3	5.2	2.9	0.1	2.8
				Middle (M)	5.0	19.4	15.1	8.1	439.3	5.9	3.4	3.5	21.3	2.1	0.4	3.1
				Outflow (O)	0.5	17.1	11.8	8.3	273.0	6.0	3.5	3.4	19.1	2.0	0.3	3.3
				Average	5.9	18.0	13.3	8.2	478.4	5.4	3.3	3.7	15.2	2.3	0.2	3.1
Juam (JA)	1010	33	108.5	Middle (M)	36.6	22.2	10.3	6.2	78.3	3.9	3.9	0.9	6.4	0.4	0.1	1.7
				Outflow (O)	38.5	22.8	8.8	6.1	70.5	3.5	2.2	0.9	8.2	0.4	0.2	1.6
				Average	37.6	22.5	9.6	6.1	74.4	3.7	2.1	0.9	7.3	0.4	0.1	1.6

## Data Availability

The data presented in this study are available on request from the corresponding author. The data are not publicly available due to reasons of privacy.

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
