# Peer review of "Transcriptional Responses of Stress-Related Genes in Pale Chub (Zacco platypus) Inhabiting Different Aquatic Environments: Application for Biomonitoring Aquatic Ecosystems"

_ijerph, 2022, doi:10.3390/ijerph191811471_

Round 1
Reviewer 1 Report
This article reports increased activation of genes involved in cellular homeostasis (HSP70; HSP90) and metal detoxification (MT) in pale chub from the lake with poor water quality. Correlations were revealed between expression of marker genes, water quality factors, and sampling sites. Therefore, stress-related genes may be useful for biomonitoring and environmental risk assessment.
The introduction provides sufficient background and includes all relevant references.
The research is well designed and illustrated.
The used methodology was adequately described.
The conclusions are supported by the results.
The manuscript may be potentially interesting, but its language should be strengthened before submission.
Please consider my comments in a pdf version. I hope it will help you to improve your manuscript.

Author Response
Reviewer # 1
This article reports increased activation of genes involved in cellular homeostasis (HSP70; HSP90) and metal detoxification (MT) in pale chub from the lake with poor water quality. Correlations were revealed between expression of marker genes, water quality factors, and sampling sites. Therefore, stress-related genes may be useful for biomonitoring and environmental risk assessment.
The introduction provides sufficient background and includes all relevant references.
The research is well designed and illustrated.
The used methodology was adequately described.
The conclusions are supported by the results.
The manuscript may be potentially interesting, but its language should be strengthened before submission.
Please consider my comments in a pdf version. I hope it will help you to improve your manuscript.
- The manuscript has been revised as reviewer 1’s comments. The revised parts were presented as the red letters in the revised manuscript as the below.
[Line 19] Lakes -> lakes
[Line 66] water -> reservoirs
[Line 69] Lakes -> lakes
[Line 80] Lakes -> lakes
[Line 138] Lakes -> lakes
[Line 156] Z. platypus -> Zacco platypus
[Line 159] Lakes -> lakes
[Line 167] Lakes -> lakes
[Line 175] Z. platypus -> Zacco platypus
[Line 178] Lakes -> lakes
[Line 197] Lakes -> lakes
[Line 202] Z. platypus (adjust line spacing)
[Line 208] Lakes -> lakes
[Line 215] Z. platypus (adjust line spacing)
[Line 223] Lakes -> lakes
[Line 225] Lakes -> lakes
[Line 228] Z. platypus -> Zacco platypus
[Line 242] Lakes -> lakes
[Line 244] Z. platypus (adjust line spacing)
[Line 260] Z. platypus (adjust line spacing)
[Line 265] Lakes -> lakes

Reviewer 2 Report
In this study, the suitability of transcriptional responses of stress-related genes in pale chub (Zacco platypus) living in different lake environments for biomonitoring applications has been demonstrated. The methodology used is appropriately defined. The findings presented are clear and discussed in accordance with the integrity of the article. The conclusions are given in line with the findings obtained. The length of the paper is acceptable. The paper is an original contribution and has not been previously published.
But there are a few points I should mention. Namely, gill and liver tissues were analyzed by qPCR using cellular homeostasis, metal detoxification, and anti-oxidation genes, but in my opinion, these can be improved. Different methods of quantification may also be required. For example, can conversion of mRNA into protein occur? Can morphological change be observed? For this, protein levels can be analyzed by western blotting. Histological sections can also be stained.
The topics I mentioned are not indispensable for a biomonitoring study. The work is already very valuable as it is. However, I suggest an environmental modeling study for Singal and Juam lakes in the next step. Therefore, I believe the results obtained from the different quantification methods mentioned above will be very useful combined with the ecological parameters.
Apart from these, I have one more suggestion in the text. In Line 57, if any, previous studies on molecular responses to environmental stressors in different species should be briefly mentioned.

Author Response
Reviewer # 2
In this study, the suitability of transcriptional responses of stress-related genes in pale chub (Zacco platypus) living in different lake environments for biomonitoring applications has been demonstrated. The methodology used is appropriately defined. The findings presented are clear and discussed in accordance with the integrity of the article. The conclusions are given in line with the findings obtained. The length of the paper is acceptable. The paper is an original contribution and has not been previously published.
But there are a few points I should mention. Namely, gill and liver tissues were analyzed by qPCR using cellular homeostasis, metal detoxification, and anti-oxidation genes, but in my opinion, these can be improved. Different methods of quantification may also be required. For example, can conversion of mRNA into protein occur? Can morphological change be observed? For this, protein levels can be analyzed by western blotting. Histological sections can also be stained.
The topics I mentioned are not indispensable for a biomonitoring study. The work is already very valuable as it is. However, I suggest an environmental modeling study for Singal and Juam lakes in the next step. Therefore, I believe the results obtained from the different quantification methods mentioned above will be very useful combined with the ecological parameters.
- Quantification at the molecular level can be used to different methods of RNA or protein work (qPCR or western blotting) for evaluating potential effects in cellular homeostasis, metal detoxification, and oxidation. The reviewer’s comment is a good point for studies of molecular responses in fish tissues according to environmental and geographical changes. However, we focus on transcriptional responses of stress-related genes in pale chub (Zacco platypus) inhabiting field lakes in the study. First of all, changes of the mRNA level were try to evaluate using cellular homeostasis, metal detoxification, and anti-oxidation genes in field fish. There are not easy to treat molecular work in fish samples collected from field lake environments. As the reviewer 3’s comments, the gene expression analysis for aquatic biomonitoring is one of the newest results, although research on biomonitoring and biomarker has been quite long developed. In addition, histological analysis is also reflected morphological changes in the tested tissues. Further study, histological and molecular (protein level) changes will be analyze in platypus for understanding different molecular responses depend on various aquatic ecosystems. Moreover, environmental modeling study for Singal and Juam lakes will be conduct in the next step.
Apart from these, I have one more suggestion in the text. In Line 57, if any, previous studies on molecular responses to environmental stressors in different species should be briefly mentioned.
- According to the comments of the reviewers, the following sentence was added to line 58 of the manuscript
[Line 58] By this time, molecular responses in fish have mostly been studied about effects to exposures of chemical compounds such as benzo(a)pyrene in Oryzias latipes or Oreochromis niloticus fish [4,16].

Reviewer 3 Report
This paper entitled Transcriptional responses of stress-related genes in pale chub (Zacco platypus) inhabiting different aquatic environments: Application for biomonitoring aquatic ecosystems was mainly focused on the correlation between water quality and gene expression.
This manuscript can be accepted for publication after addressing the following minor comments.
1. Research on biomonitoring and biomarker has been quite long developed; however, the gene expression approach is probably the newest.
2. The differences between biomonitor and biomarker could be explained more clearly, supported by the newest references
3. The novelty of this research able to be improved, which makes it differ from similar research
4. The other important thing that can develop is using native fish species as a biomarker of water quality. This could be discussed more deeply as a novelty
5. What is the implication of this research ? Using this native species for biomonitoring water quality?
Author Response
Reviewer # 3
This paper entitled Transcriptional responses of stress-related genes in pale chub (Zacco platypus) inhabiting different aquatic environments: Application for biomonitoring aquatic ecosystems was mainly focused on the correlation between water quality and gene expression.
This manuscript can be accepted for publication after addressing the following minor comments.
1.Research on biomonitoring and biomarker has been quite long developed; however, the gene expression approach is probably the newest.
- Thank you for reviewer’s comments. These results are one of the newest approaches.
- The differences between biomonitor and biomarker could be explained more clearly, supported by the newest references
- Biomarkers are described as checkpoints that reflect the varying levels of changes in organisms, tissues and cells. Biomonitor is the measurement of biological health through as comprehensive understanding of individual biomarkers. Based on the comment of the reviewer, the manuscript has been revised as follows.
[Line 52] In addition, biomonitoring through a comprehensive understanding of biomarkers can represent measures of health in the freshwater environments.
- The novelty of this research able to be improved, which makes it differ from similar research
- According to the comment of the reviewer, a sentence has been added to the introduction about the differentiation of this study.
[Line 60] The previous reported studies on molecular markers were mostly limited to research through toxicity test in the laboratory, using experimental rearing species that are relatively easy to handling.
- The other important thing that can develop is using native fish species as a biomarker of water quality. This could be discussed more deeply as a novelty
- According to the comment of the reviewer, the advantages of biomarker research using platypus distributed throughout South Korea rivers were described in the discussion.
[Line 232] In addition, the molecular biomarker approach using Z. platypus, which is distributed over a wide range in rivers, is a useful tool that can represent the health of the fish habitat environment using native fish.
- What is the implication of this research? Using this native species for biomonitoring water quality?
- We mentioned in the last sentence of the conclusion as follow “Transcriptional responses using native fish platypus for biomonitoring can indicate stressful water environments in ecosystems. Our results indicated that upregulated response of HSP70, HSP90, and MT at mRNA level can reflect stressful conditions in cellular homeostasis and metal detoxification of Z. platypus fish inhabiting different lake environments. This research will use to application for assessing health conditions of aquatic environments as well as live organisms such as fish
